# Deep Actor-Critics with Tight Risk Certificates

## Abstract

After a period of research, deep actor-critic algorithms have reached a level where they influence our everyday lives. They serve as the driving force behind the continual improvement of large language models through user-collected feedback. However, their deployment in physical systems is not yet widely adopted, mainly because no validation scheme that quantifies their risk of malfunction. We demonstrate that it is possible to develop tight risk certificates for deep actor-critic algorithms that predict generalization performance from validation-time observations. Our key insight centers on the effectiveness of minimal evaluation data. Surprisingly, a small feasible of evaluation roll-outs collected from a pretrained policy suffices to produce accurate risk certificates when combined with a simple adaptation of PAC-Bayes theory. Specifically, we adopt a recently introduced recursive PAC-Bayes approach, which splits validation data into portions and recursively builds PAC-Bayes bounds on the excess loss of each portion's predictor, using the predictor from the previous portion as a data-informed prior. Our empirical results across multiple locomotion tasks and policy expertise levels demonstrate risk certificates that are tight enough to be considered for practical use.

## 1 Introduction

Reinforcement Learning (RL) is transforming emerging AI technologies. Large language models incorporate human feedback via RL, thereby continually improving their accuracy [Christiano et al., 2017, Ziegler et al., 2019, DeepSeek-AI et al., 2025]. Generative AI is increasingly being integrated into agentic workflows to automate complex decision making tasks. RL has also shown great promise in the control of physical robotic systems. Recent deep actor-critic algorithms learned to make a legged robot walk after only 20 minutes of outdoor training in an online mode [Kostrikov et al., 2023]. Model-based extensions of actor-critic pipelines can also achieve sample-efficient visual-control tasks in diverse settings [Hafner et al., 2025, Zhang et al., 2023]. Despite the exciting results observed in experimental conditions, RL is used far less than classical approaches in physical robot control. This opportunity has largely been missed mainly because deep RL algorithms are overly sensitive to initial conditions and can change behavior drastically during training. Embodied intelligent systems have a high risk of causing harm when their generalization performance differs significantly from their observed validation performance. Predictable generalization performance is even more critical when these systems update their behavior based on interactions with humans.

There has been an effort to use learning-theoretic approaches to train high-capacity predictors with risk certificates, i.e., bounds that guarantee a predictor's generalization performance. Typically, this performance is estimated from observed validation results, which may be misleading. *Probably Approximately Correct Bayesian (PAC-Bayes) theory* [McAllester, 1999, Alquier et al., 2024] provides risk certificates for stochastic predictors, relative to a prior distribution over the hypothesis space. In this framework, the computationally prohibitive capacity term is reduced to a Kullback-Leibler

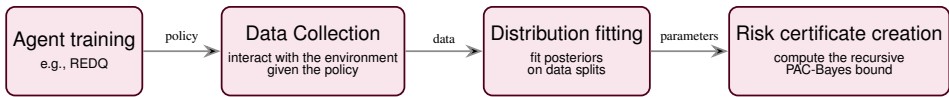

Figure 1: Our four steps to generate tight risk certificates for deep actor-critic algorithms.

divergence between the posterior and the prior, enabling the incorporation of domain knowledge into the analysis. Since we often deal with stochastic policies, relying on PAC-Bayes is a natural choice.

PAC-Bayes is the first and remains the most promising method for providing meaningful risk certificates to deep neural networks [Dziugaite and Roy, 2017, Pérez-Ortiz et al., 2021, Lotfi et al., 2022]. Further studies have improved the tightness, i.e., precision, of these certificates through the following techniques: (i) pretraining probabilistic neural nets on held-out data and using them as *data-informed priors* [Ambroladze et al., 2006, Dziugaite et al., 2021]; (ii) using pretrained networks as first-step predictors and developing PAC-Bayes guarantees on the residual of their predictions, termed the *excess loss*; and (iii) recursively repeating the first two steps on multiple data splits, a recent method known as the *Recursive PAC-Bayes* [Wu et al., 2024]. The scope of these exciting developments has thus far been limited to simple classification tasks with feedforward neural networks. Their application to deep actor-critic algorithms remains open, primarily because the mainstream PAC-Bayes bounds assume i.i.d. datasets, whereas RL assumes a controlled Markov chain.

We present a simple recipe for providing risk certificates for deep model-free actor-critic architectures. We find that, contrary to what one might expect, the three modern PAC-Bayesian learning tools mentioned above can successfully handle the high variance of Monte Carlo samples collected by running a pretrained policy network for multiple episodes in evaluation mode. Our approach proposes self-certified training of probabilistic neural networks on different splits of an i.i.d. data set containing return realizations of the policy, computed by first-visit Monte Carlo and post-processed through a simple thinning approach. We recursively build a PAC-Bayes bound on the excess losses of these networks, following a new adaptation of the recipe introduced by Wu et al. [2024]. Figure 1 illustrates our risk-certificate generation workflow. Our results highlight that the risk certificates get significantly tighter as the recursion depth increases. The final bounds are tight enough for practical use. Furthermore, the tightness of the risk certificates is proportional to the policy's level of expertise.

## 2 Background

### 2.1 The state of the art of model-free deep actor-critic learning

Consider a set of states $\mathcal{S}$ an agent may be in and an action space $\mathcal{A}$ from which the agent can choose actions to interact with its environment. Denote by $\Delta(\mathcal{S})$ and $\Delta(\mathcal{A})$ the sets of probability distributions defined on $\mathcal{S}$ and $\mathcal{A}$, respectively. We define a Markov Decision Process (MDP) [Puterman, 2014] as the tuple $M = \langle \mathcal{S}, \mathcal{A}, r, P, P_0, \gamma \rangle$, where $r : \mathcal{S} \times \mathcal{A} \to [0, R]$ is a bounded reward function, $P : \mathcal{S} \times \mathcal{A} \times \mathcal{S} \to [0, 1]$ is the state-transition kernel conditioned on a state-action pair; specifically $P(s'|s, a)$ is the probability distribution of the next state $s' \in \mathcal{S}$ given the current state-action pair $(s, a) \in \mathcal{S} \times \mathcal{A}$. We denote the initial-state distribution by $P_0 \in \Delta(\mathcal{S})$, the discount factor by $\gamma \in (0, 1)$, and let $\pi : \mathcal{S} \times \mathcal{A} \to [0, 1]$ be a policy. The goal of RL is to learn a policy that maximizes the expected discounted return, $\pi_* := \arg\max_{\pi \in \Pi} \mathbb{E}_{\tau_\pi} \left[ \sum_{t=0}^{\infty} \gamma^t r(s_t, a_t) \right]$. The expectation is taken with respect to the trajectory $\tau_\pi := (s_0, a_0, s_1, a_1, s_2, a_2, \ldots)$ of states and actions generated when a policy $\pi$ chosen from a feasible set $\Pi$ is executed. We refer to $\pi_*$ as the optimal policy. The exact Bellman operator for a policy $\pi$ is defined as

$$T_\pi Q(s, a) := r(s, a) + \gamma \mathbb{E}_{s' \sim P(\cdot|s,a)} \left[ Q(s', \pi(s')) \right] \tag{1}$$

for some function $Q : \mathcal{S} \times \mathcal{A} \to \mathbb{R}$. The unique fixed point of this operator is the true action-value function $Q_\pi$, which maps a state-action pair $(s, a)$ to the expected discounted sum of rewards the policy $\pi$ collects when executed from $(s, a)$. In other words, the equality $T_\pi Q(s, a) = Q(s, a)$ holds if and only if $Q(s, a) = Q_\pi(s, a), \forall (s, a)$. Any other $Q$ incurs an error $(T_\pi Q(s, a) - Q(s, a))^2$, called the *Bellman error*. Common deep actor-critic methods approximate the true action-value function $Q_\pi$ by one-step Temporal Difference (TD) learning that minimizes $L(Q, \pi) := \mathbb{E}_{s \sim P_\pi}[(T_\pi Q(s, a) - Q(s, a))^2]$ with respect to $Q$, given a data set $\mathcal{D}$ and $P_\pi(s' \in A) = \mathbb{E}_{s \sim P_0} \left[ \sum_{t>0} P(s_t \in A|s_0 = s, \pi(s)) \right]$ which is defined as the state-visitation distribution of policy $\pi$ for some event $A$ that belongs to the $\sigma$-algebra of the transition probability

distribution. Because the transition probabilities are unknown, the expectation term in Eq. 1 cannot be computed. Instead, the observed transitions are used to approximate it with a single-sample Monte Carlo estimate, yielding the training objective below:

$$\widetilde{L}(Q) := \mathbb{E}_{s \sim P_\pi} \Big[ \mathbb{E}_{s' \sim P(\cdot | s, \pi(s))} \left[ (r(s,a) + \gamma Q(s', \pi(s')) - Q(s,a))^2 \right] \Big].$$

A deep actor-critic algorithm fits a neural-network function approximator $Q$, referred to as the critic, to a set of observed tuples $(s, a, s')$ stored in a replay buffer $\mathcal{D}$ by minimizing an empirical estimate of the stochastic loss: $\widehat{L}_{\mathcal{D}}(Q) := 1/|\mathcal{D}| \sum_{(s,a,s') \in \mathcal{D}} (\widehat{T}_\pi Q(s,a,s') - Q(s,a))^2$. The critic is then used to train a policy network, or actor, $\pi' \leftarrow \arg\max_\pi \mathbb{E}_{s \sim P_\pi} [Q(s, \pi(s))]$. It is common practice to adopt the *Maximum-Entropy Reinforcement Learning* approach [Haarnoja et al., 2018a,b] to balance exploration and exploitation, thereby ensuring effective training. The approach supplements the reward function with a policy-entropy term $r_{\text{MaxEnt}}(s,a) = r(s,a) + \alpha \mathbb{H}[\pi(\cdot|s)]$, where $\alpha \geq 0$ is a scaling hyperparameter tuned jointly with the actor and critic.

Performing off-policy TD learning with deep neural nets is notoriously unstable which is often attributed to the *deadly triad* [Sutton and Barto, 2018]. The main source of instability is the accumulation of errors from approximating $T_\pi Q$ by its Monte Carlo estimate. Strategies to improve stability include maintaining Polyak-updated target networks [Lillicrap et al., 2016] and learning twin critics while using the minimum of their target-network outputs in Bellman target calculation [Fujimoto et al., 2018]. Empirically, training an ensemble of critic networks in a maximum-entropy setup largely mitigates these stability issues. We adopt REDQ [Chen et al., 2021], a state-of-the-art actor-critic method for model-free continuous control, as our representative approach. This choice is pragmatic rather than restrictive allowing us to trade the computational cost of a broader exploration of algorithms for a deeper, more comprehensive empirical evaluation of a single one.

## 2.2 Developing risk certificates with PAC-Bayes bounds

PAC-Bayes [McAllester, 1999, Alquier et al., 2024] offers a powerful way to understand and control how well learning algorithms generalize by blending prior beliefs with what we learn from data. *PAC-Bayesian learning* uses modern machine learning techniques to model $\rho$ with complex function approximators and fit them to data. It has been successfully applied in both image classification [Dziugaite and Roy, 2017, Wu et al., 2024] and regression tasks [Reeb et al., 2018]. Its application to reinforcement learning has so far been limited to the design of critic training losses without rigorously quantifying the tightness of the performance guarantees [Tasdighi et al., 2024a,b].

**Notation.** Let $\mathcal{H} : \mathcal{X} \rightarrow \mathcal{Y}$ be a set of feasible hypotheses and $\ell : \mathcal{Y} \times \mathcal{Y} \rightarrow [0, 1]$ be a bounded loss function.[1] Further, let $L(h) = \mathbb{E}_{(x,y) \sim P_D} [\ell(h(x), y)]$ be the expected error, where $P_D$ is a distribution on $\mathcal{X} \times \mathcal{Y}$. The empirical loss is $\hat{L}(h) = \frac{1}{N} \sum_{i=1}^{N} \ell(h(x_i), y_i)$ for a data set $\mathcal{D} = \{(x_n, y_n) : n \in \{1, \ldots, N\}\}$ of size $N$ with $(x_n, y_n) \sim P_D$. $\mathcal{P}$ is the set of distributions on $\mathcal{H}$. For two distributions $\rho, \rho_0$ on $\mathcal{H}$, the Kullback-Leibler (KL) divergence is defined as $\text{KL}(\rho \| \rho_0) \triangleq \mathbb{E}_{h \sim \rho} [\log \rho(h) - \log \rho_0(h)]$. We use $\text{kl}(p \| q) \triangleq p \log(p/q) + (1-p) \log((1-p)/(1-q))$ to denote the KL divergence between two Bernoulli distributions. PAC-Bayesian analysis [McAllester, 1999, Shawe-Taylor and Williamson, 1997, Alquier et al., 2024] develops bounds on the *expected loss* $\mathbb{E}_{h \sim \rho} [L(h)]$, under a posterior distribution $\rho$ with respect to a prior distribution $\rho_0$, that hold with high probability. That is, they provide *risk certificates* for the generalization error. For brevity, we will use $\mathbb{E}_\rho [\cdot] = \mathbb{E}_{h \sim \rho} [\cdot]$ throughout this paper. In the context of PAC-Bayes, the terms *posterior* and *prior* refer to distributions dependent and independent of the validation data, respectively. They are not to be understood in a Bayesian manner as being linked by a likelihood.[2] Which bounds one should choose to get the tightest risk certificates depends on the specific use case; see, e.g., Alquier et al. [2024] for a recent introduction and a survey of various bounds. In this work we rely on bounds derived from the kl divergence as they are tighter than the alternatives when no additional information about the data distribution is available, while noting that the same arguments apply to any other PAC-Bayesian bound.

### 2.2.1 PAC-Bayes-kl bound

Assuming the definitions given above, the *PAC-Bayes-kl bound* is given by

---

[1]Our discussion generalizes directly to any bounded loss within an interval $[a, b]$ with $a, b \in \mathbb{R}$.

[2]See Germain et al. [2016] for results linking PAC-Bayes and Bayesian inference.

**Theorem 2.1** (PAC-Bayes-kl bound [Seeger, 2002, Maurer, 2004]). *For any probability distribution $\rho_0 \in \mathcal{P}$ that is independent of $\mathcal{D}$ and any $\delta \in (0,1)$, we have*

$$\mathbb{P}\left(\exists \rho \in \mathcal{P} : \mathrm{kl}\big(\mathbb{E}_\rho[\hat{L}(h)]||\mathbb{E}_\rho[L(h)]\big) \geq \big(\mathrm{KL}\left(\rho \parallel \rho_0\right) + \ln(2\sqrt{N}/\delta)\big)/N\right) \leq \delta.$$

*Proof.* See, e.g., Maurer [2004] for a proof of the bound. $\qquad\square$

We define the upper inverse of $\mathrm{kl}\left(\cdot \parallel \cdot\right)$ as $\mathrm{kl}^{-1,+}(\hat{p}, \varepsilon) \triangleq \max\{p : p \in [0,1] \mid \mathrm{kl}\left(\hat{p} \parallel p\right) \leq \varepsilon\}$ and the lower one as $\mathrm{kl}^{-1,-}(\hat{p}, \varepsilon) \triangleq \min\{p : p \in [0,1], \mathrm{kl}\left(\hat{p} \parallel p\right) \leq \varepsilon\}$ and cite the following inequality.

**Lemma 2.2** (kl-inequality [Langford, 2005, Foong et al., 2021, 2022]). *Let $Z_1, \ldots, Z_N$ be i.i.d. random variables taking values on an interval $[0,1]$ and $\mathbb{E}[Z_n] = p$ for all $n$. Let their empirical mean be $\hat{p} = \frac{1}{N}\sum_{n=1}^N Z_n$. Then, for any $\delta \in (0,1)$ we have*

$$\mathbb{P}\left(\mathrm{kl}(\hat{p}||p) \geq \ln(1/\delta)/N\right) \leq \delta,$$

*the inverse of which is given by*

$$\mathbb{P}\left(p \geq \mathrm{kl}^{-1,+}\left(\hat{p}, \ln(1/\delta)/N\right)\right) \leq \delta, \quad and \quad \mathbb{P}\left(p \leq \mathrm{kl}^{-1,-}\left(\hat{p}, \ln(1/\delta)/N\right)\right) \leq \delta.$$

*Proof.* See Langford [2005], Corollary 3.7 for a proof of the bound. $\qquad\square$

### 2.2.2 PAC-Bayes-Split-kl bound

Wu and Seldin [2022] generalize these bounds to random variables that take values in intervals $[a, b]$ splitting each into two components that individually satisfy the constraints of the kl-inequality.

Let $Z \in [a, b]$, with $a, b \in \mathbb{R}$, be a random variable and set $p = \mathbb{E}[Z]$. For $\mu \in [a, b]$ define $Z^+ = \max\{0, Z - \mu\}$ and $Z^- = \max\{0, \mu - Z\}$, so that $Z = \mu + Z^+ - Z^-$. Let $p^+ = \mathbb{E}[Z^+]$ and $p^- = \mathbb{E}[Z^-]$ be their respective expectations, and let $\hat{p}^+ = \frac{1}{N}\sum_{n=1}^N Z_n^+$ and $\hat{p}^- = \frac{1}{N}\sum_{n=1}^N Z_n^-$ be their empirical means for an i.i.d. sample $Z_1, \ldots, Z_N$. The *split*-kl *inequality* is stated below.

**Lemma 2.3** (Split-kl inequality [Wu and Seldin, 2022]). *For any $\mu \in [a, b]$ and $\delta \in (0,1)$*

$$\mathbb{P}\left(p \leq \mu + (b - \mu)\mathrm{kl}^{-1,+}\left(\frac{\hat{p}^+}{b - \mu}, \frac{\ln(2/\delta)}{N}\right) - (\mu - a)\mathrm{kl}^{-1,-}\left(\frac{\hat{p}^-}{\mu - a}, \frac{\ln(2/\delta)}{N}\right)\right) \geq 1 - \delta.$$

*Proof.* The lemma follows by applying Lemma 2.2 to each of the kl terms and a union bound. $\qquad\square$

For the PAC-Bayesian analogue, define $\tilde{\ell} : \mathcal{Y} \times \mathcal{Y} \to [a, b]$, where $a, b \in \mathbb{R}$. For $\mu \in [a, b]$, define $\tilde{\ell}^+ = \max\{0, \tilde{\ell} - \mu\}$ and $\tilde{\ell}^- = \max\{0, \mu - \tilde{\ell}\}$. $\tilde{L}^+(h) = \mathbb{E}_{(x,y) \sim P_D}[\tilde{\ell}^+(h(x), y)]$ and $\hat{\tilde{L}}^+(h) = \frac{1}{N}\sum_{n=1}^N \tilde{\ell}^+(h(x_n), y_n)$ are the expected and empirical losses. $L^-$ and $\hat{\tilde{L}}^-$ are defined analogously. With these definitions, we now cite the PAC-Bayes-split-kl inequality.

**Theorem 2.4** (PAC-Bayes-Split-kl inequality [Wu and Seldin, 2022]). *Let $\tilde{\ell}$ and the remaining loss terms be defined as above. Then for any $\rho_0$ on $\mathcal{H}$ independent of $\mathcal{D}$, any $\mu \in [a, b]$, and any $\delta \in (0,1)$*

$$\mathbb{P}\left(\exists \rho \in \mathcal{P} : \mathbb{E}_\rho[\tilde{L}(h)] \geq \mu + (b - \mu)\mathrm{kl}^{-1,+}\left(\frac{\mathbb{E}_\rho[\hat{\tilde{L}}^+(h)]}{b - \mu}, \frac{\mathrm{KL}\left(\rho \parallel \rho_0\right) + \ln(4\sqrt{N}/\delta)}{N}\right)\right.$$
$$\left. - (\mu - a)\mathrm{kl}^{-1,+}\left(\frac{\mathbb{E}_\rho[\hat{\tilde{L}}^-(h)]}{\mu - a}, \frac{\mathrm{KL}\left(\rho \parallel \rho_0\right) + \ln(4\sqrt{N}/\delta)}{N}\right)\right) \leq \delta.$$

*Proof.* The theorem follows by applying Lemma 2.3 to the decomposition $\mathbb{E}_\rho[\tilde{L}(h)] = \mu + \mathbb{E}_\rho[\tilde{L}^+(h)] - \mathbb{E}_\rho[\tilde{L}^-(h)]$. $\qquad\square$

### 2.2.3 Recursive PAC-Bayes bound

**Data-informed prior.** The tightness of PAC-Bayesian bounds is dominated by the KL divergence between the posterior $\rho$ and the prior $\rho_0$. The better the prior guess is, the tighter the bound. Because the prior must be be independent of the observed data, a common choice is to select a prior that is as uniform as possible over the hypothesis space. To improve upon this naïve choice, Ambroladze et al. [2006] proposed splitting the observed data into two disjoint subsets $S_0$ and $S_1$, i.e., $\mathcal{D} = S_0 \cup S_1$, using $S_0$ to infer a *data-informed prior* and $S_1$ to subsequently evaluate the bound. This approach balances the benefit of a better prior with the cost of having fewer observations to evaluate the bound.

**Excess loss.** The *excess loss* $L^{\text{exc}}(h)$ with respect to a reference hypothesis $h^* \in \mathcal{H}$ is defined as $L^{\text{exc}}(h) = L(h) - L(h^*)$. The excess-loss concept allows us to decompose the expected loss as $\mathbb{E}_\rho [L(h)] = \mathbb{E}_\rho [L(h) - L(h^*)] + L(h^*)$. Using $S_0$ to construct both the prior $\rho_0$ and the reference $h^*$, Mhammedi et al. [2019] showed that, assuming $L(h^*)$ is close to $L(h)$, the excess loss has lower variance and thus yields a more efficient bound, while a bound on $L(h^*)$ is independent of $\text{KL}(\rho \parallel \rho_0)$ and can be obtained using standard generalization guarantees.

**Recursive PAC-Bayes.** Wu et al. [2024] generalized the excess loss further by introducing a scaling factor $\kappa < 1$ to maintain a diminishing effect of recursions: $\mathbb{E}_\rho [L(h)] = \mathbb{E}_\rho [L(h) - \kappa \mathbb{E}_{\rho_0} [L(h^*)]] + \kappa \mathbb{E}_{\rho_0} [L(h^*)]$. Here, the first term reflects the excess loss with respect to a scaled version of the expected reference hypothesis loss under the prior $\rho_0$. The second term in turn is an expected loss again similar to the one on the left-hand side of the equation. Instead of adhering to a binary split $\mathcal{D} = S_0 \cup S_1$ such that $S_0 \cap S_1 = \emptyset$, they propose to extend this decomposition recursively, by partitioning $\mathcal{D}$ into $T$ disjoint subsets, $\mathcal{D} = \bigcup_{t=1}^{T} S_t$ and and they define $S_{\leq t} = \bigcup_{s=1}^{t} S_s$ and $S_{\geq t} = \bigcup_{s=t}^{T} S_s$. Their recursion is given by

$$\mathbb{E}_{\rho_t} [L(h)] = \mathbb{E}_{\rho_t} [L(h) - \kappa_t \mathbb{E}_{\rho_{t-1}} [L(h)]] + \kappa_t \mathbb{E}_{\rho_{t-1}} [L(h)], \tag{2}$$

for $t \geq 2$, and $\kappa_1, \ldots, \kappa_T$ are scaling factors. The distributions $\rho_1, \ldots, \rho_T \in \mathcal{H}$ form a sequence such that $\rho_t$ depends solely on $S_{\leq t}$ and $S_{\geq t}$ to estimate $\mathbb{E}_{\rho_t} [L(h)]$.

While Wu et al. [2024] formulate their final recursive bound directly for a zero-one loss and PAC-Bayes split-kl bounds [Wu and Seldin, 2022], we present their result first in a general loss-agnostic form before we construct a specific bound in the next section.

**Theorem 2.5.** *(Recursive PAC-Bayes bound.) Let $\mathcal{D} = S_1 \cup \cdots \cup S_T$ be a disjoint decomposition of the set of observations $\mathcal{D}$. Let $S_{\leq t}$ and $S_{\geq t}$ be as defined above, $N = |\mathcal{D}|$, and $N_t = |S_{\geq t}|$. Let $\kappa_1, \ldots, \kappa_T$ be a sequence of scaling factors, where $\kappa_t$ is allowed to depend on $S_{\leq t-1}$. Let $\mathcal{P}_t$ be the set of distributions on $\mathcal{H}$ which are allowed to depend on $S_{\leq t}$, and $\rho_t \in \mathcal{P}_t$. Then, for any $\delta \in (0, 1)$,*

$$\mathbb{P}\left( \exists t \in [T], \rho_t \in \mathcal{P}_t \text{ such that } \mathbb{E}_{\rho_t} [L(h)] \geq \mathcal{B}_t(\rho_t) \right) \leq \delta,$$

*where $\mathcal{B}_t(\rho_t)$ is a generic PAC-Bayesian bound on $\mathbb{E}_{\rho_t} [L(h)]$ defined recursively as follows.*

$$\mathcal{B}_t(\rho_t) = \mathcal{E}_t(\rho_t, \kappa_t) + \kappa_t \mathcal{B}_{t-1}(\rho_{t-1}^*),$$

*where $\mathcal{B}_1(\rho_1)$ is a PAC-Bayes bound on $\mathbb{E}_{\rho_1} [L(h)]$ with an uninformed prior and $\mathcal{E}_t(\rho_t, \kappa_t)$ is a PAC-Bayes bound on the excess loss $\mathbb{E}_{\rho_t} \left[ L(h) - \kappa_t \mathbb{E}_{\rho_{t-1}^*} [L(h')] \right]$.*

*Proof.* Because $\mathcal{B}_1(\rho_1)$ and $\mathcal{E}_t(\rho_t, \kappa_t)$ are PAC-Bayes bounds by assumption, we have

$$\mathbb{P}\left( \exists \rho_1 \in \mathcal{P}_1 : \mathbb{E}_{\rho_1} [L(h)] \geq \mathcal{B}_1(\rho_1) \right) \leq \delta/T,$$

$$\text{and} \quad \mathbb{P}\left( \exists \rho_t \in \mathcal{P}_t : \mathbb{E}_{\rho_t} \left[ L(h) - \kappa_t \mathbb{E}_{\rho_{t-1}^*} [L(h')] \right] \geq \mathcal{E}_t(\rho_t, \kappa_t) \right) \leq \delta/T \text{ for } t \in \{2, \ldots, T\}.$$

The claim follows by expected loss decomposition and the recursion. $\qquad\square$

## 3 Recursive PAC-Bayesian risk certificates for reinforcement learning

Obtaining risk certificates involves four steps, following our conceptual structure in Figure 1.

*(i) Training an agent.* The chosen actor-critic algorithm, REDQ [Chen et al., 2021], which we use in our experiments, is trained until convergence or until a computational budget is exhausted, after which we freeze its policy parameters, e.g., the weights of the corresponding neural net.

*(ii) Collecting data.* After training the policy, we run an agent acting according to this policy for several episodes. Although a PAC-Bayesian bound gets tighter as the number of data points increases, we observe that even a relatively small number of evaluation roll-outs is sufficient to get tight results.

*(iii) Fitting the posteriors.* We rely on the discounted return as the prediction target rather than a plain sum of rewards for several reasons. Short-term risks tend to be more relevant for decisions, as longer-term risks depend on an increasing set of external, usually unaccountable, factors. Discounted rewards also serve as a proxy for lifelong learning and policy evaluation as they generalize to non-episodic data. That said, even though the original policy might be trained on discounted returns in step *(i)*, a valid bound could also be constructed by computing the non-discounted rewards from data collected in *(ii)*. As discussed in Section 2.2.3, we split the data into $T$ disjoint subsets and train a series of $T$ last-layer Bayesian neural nets via first-visit Monte Carlo to infer distributions over $S_{\leq t}$.

*(iv) Construction of the bound.* As discussed above, we focus on a generally well-performing set of kl-based bounds. We construct the following bounds for $\mathcal{B}_1$ and $\mathcal{E}_t$ ($t \in \{1, \ldots, T\}$).

**A bound for $\mathcal{B}_1$.** As $\hat{L}(h)$ is bounded between $[0, B]$, we rescale its expectation and choose

$$\mathcal{B}_1(\rho_1) = B\mathrm{kl}^{-1,+}\left( \frac{\mathbb{E}_\rho[\hat{L}(h)]}{B}, \frac{\mathrm{KL}\left(\rho_1 \parallel \rho_0^*\right) + \ln(2T\sqrt{n}/\delta)}{N} \right),$$

where $\rho_0^*$ is a data-independent prior distribution on $\mathcal{H}$. Given the result in Theorem 2.1, this is a PAC-Bayesian bound on $\mathbb{E}_{\rho_1}[L(h)]$, i.e., $\mathbb{P}\big(\exists \rho_1 \in \mathcal{P}_1 : \mathbb{E}_{\rho_1}[L(h)] \geq \mathcal{B}_1(\rho_1)\big) \leq \delta/T$.

**A bound for $\mathcal{E}_t$.** Let $L_t^{\mathrm{exc}}(h) = L(h) - \kappa_t \mathbb{E}_{\rho_{t-1}}[L(h')] \in [-\kappa_t B, B]$. For $\mu \in [-\kappa_t B, B]$, define $L_t^{\mathrm{exc}+}(h) = \max\{0, L_t^{\mathrm{exc}}(h) - \mu\}$ and $L_t^{\mathrm{exc}-}(h) = \max\{0, \mu - L_t^{\mathrm{exc}}(h)\}$, with $\hat{L}_t^{\mathrm{exc}+}(h)$ and $\hat{L}_t^{\mathrm{exc}-}(h)$ as their empirical analogous. We set

$$\mathcal{E}_t(\rho_t) = \mu + (B - \mu)\mathrm{kl}^{-1,+}\left( \frac{\mathbb{E}_{\rho_t}[\hat{L}_t^{\mathrm{exc}+}(h)]}{B - \mu}, \frac{\mathrm{KL}\left(\rho_t \parallel \rho_{t-1}^*\right) + \ln(4T\sqrt{N_t}/\delta)}{N_t} \right)$$
$$- (\mu + \kappa_t B)\mathrm{kl}^{-1,+}\left( \frac{\mathbb{E}_{\rho_t}[\hat{L}_t^{\mathrm{exc}-}(h)]}{\mu + \kappa_t B}, \frac{\mathrm{KL}\left(\rho_t \parallel \rho_{t-1}^*\right) + \ln(4T\sqrt{N_t}/\delta)}{N_t} \right),$$

where $\rho_{t-1}^*$ is a distribution on $\mathcal{H}$ informed by $S_{\leq t-1}$. Via Theorem 2.4 this is a PAC-Bayesian bound on $\mathbb{E}_{\rho_t}[L_t^{\mathrm{exc}}]$ that holds with a probability greater than $1 - \delta/T$, i.e.,

$$\mathbb{P}\big(\exists \rho_t \in \mathcal{P}_t \text{ such that } \mathbb{E}_{\rho_t}[L_t^{\mathrm{exc}}(h)] \geq \mathcal{E}_t(\rho_t)\big) \leq \delta/T.$$

Applying this construction recursively with $T$ steps therefore gives us a recursive PAC-Bayesian bound that holds with probability greater than $1 - \delta$.

# 4 Experiments

We perform experiments to answer the following three questions: (**Q1**) Can the test-time return of a policy $\pi$ be predicted with high precision across a range of environments and policies of varying expertise? (**Q2**) What is the influence of a PAC-Bayes bound's structure? (**Q3**) How does the validation set size influence the tightness of the risk certificate guarantee?

## 4.1 Experiment design

To evaluate our certificate-generation pipeline at an error tolerance of $\delta = 0.05$, we choose REDQ [Chen et al., 2021] as a representative state-of-the-art, sample-efficient, model-free continuous control algorithm. All REDQ hyperparameters follow those in the original paper. We first train a REDQ agent for 300 000 steps using an ensemble of ten critics, randomly sampling two at each Bellman-target evaluation for min-clipping. The learned policy is then run in evaluation mode for 100 episodes. The resulting state transitions and rewards are stored as the data set used for bound fitting. Subsequently, we run the trained policy for another 100 episodes to obtain a test dataset to compute a proxy for the generalization performance. We predict the discounted return of the policy on the test set by fitting a PAC-Bayes bound using observations from the validation set.

**Bound vs Test Error**

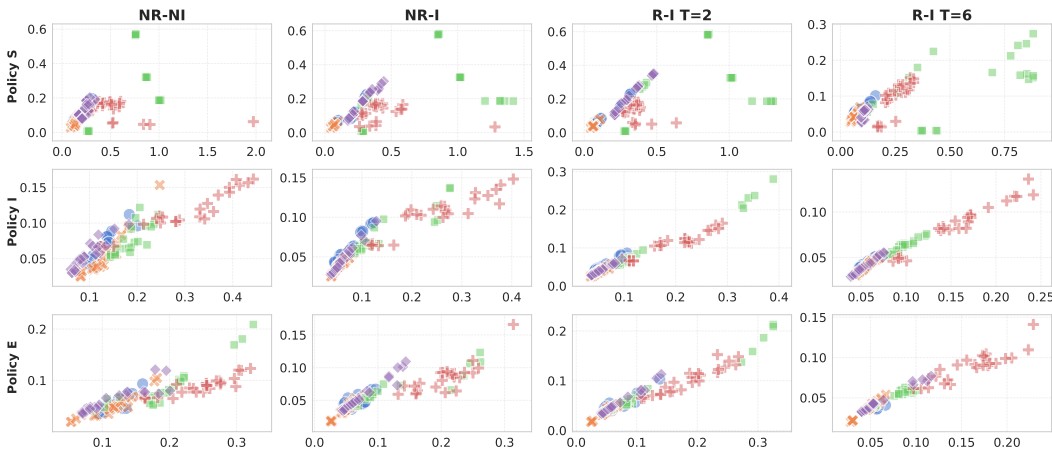

Figure 2: *Correlation plots.* PAC-Bayes bounds, one in each column, are plotted on the x-axis against true test errors on the y-axis for each method across all environments, policy instances, and repetitions to visualize correlation. Environments are color-coded as follows: `Ant` (blue circle), `Half-Cheetah` (orange cross), `Hopper` (green square), `Humanoid` (red plus), and `Walker2d` (purple diamond).

We evaluate and compare the final posterior loss $\rho$ on the full training data, and on the held-out test data, alongside the corresponding PAC-Bayes bounds across all methods and environments. To mitigate the overfitting common in continuous-control settings, where consecutive samples are highly correlated, we apply a thinning strategy that reduces redundancy while preserving data diversity. Full details on each experiment are provided in Appendix D. We provide an implementation at `anonymous`.

**Policy instances.** We define a policy instance as the output of a single policy-training round. In our experiments, we consider five policy instances, each obtained by running the REDQ algorithm with a different initial seed. Due to the stochastic nature of initialization and training, each instance follows a unique trajectory. We construct individual bounds for each instance and report them in Appendix D. To account for randomness in the risk certificate generation process, we repeat the procedure five times for every policy instance. To address question (Q2), we create separate risk certificates for three training stages of each policy, each reflecting a different level of expertise: *Starter (S)* for a policy trained for 100 000 steps, *Intermediate (I)* for 200 000 steps, and *Expert (E)* for 300 000 steps, after which no performance improvement observed.

**Environments.** We evaluate five MuJoCo environments: `Ant`, `Half-Cheetah`, `Hopper`, `Humanoid`, and `Walker2d` [Todorov et al., 2012] due to their widespread use in the community and the representative value of the platforms for real-world use cases. Risk certificates may be particularly interesting for mobile platforms that interact with their surroundings as well as humans.

**Baselines.** We design our baselines with the following points in mind: 1. how well a PAC-Bayes bound predicts test-time performance, 2. whether informative priors yield tighter guarantees, 3. whether the bound gets tighter when the recursive scheme is used, and 4. whether increasing the recursion depth improve tightness. As this is the first work to evaluate generalization bounds tailored for continuous control with deep actor-critics, there are no existing baselines for comparison. We consider two non-recursive (NR) baselines: *non-informed (NR-NI)*, a PAC-Bayes-kl inverse bound (see Theorem 2.1) with a non-informative prior that is independent of the training data, and *informed (NR-I)*, a data-informed variant in which the dataset is split equally into $\mathcal{D} = \mathcal{D}_{\text{prior}} \cup \mathcal{D}_{\text{bound}}$, allowing the prior to depend on $\mathcal{D}_{\text{prior}}$ and the empirical loss to be computed on $\mathcal{D}_{\text{bound}}$. We evaluate two recursion (R) depths, *depth two (R-I T=2)* and *depth six (R-I T=6)*, to test the effect of recursion.

**Performance metrics.** We evaluate the bounds based on three metrics: *Normalized bound value*: To ensure comparability across environments with different reward scales, we normalize the squared discounted return prediction errors by the maximum observed return during training. A value close to

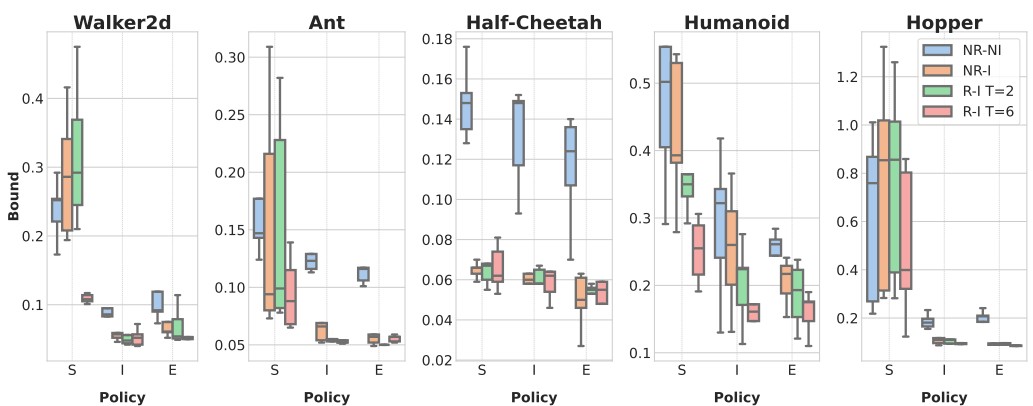

Figure 3: *Bound values.* Normalized bound values for all baselines across five MuJoCo environments over three policy qualities. Results are aggregated over all policy instances and repetitions.

zero implies that the bound closely follows the actual returns. *Tightness*: The difference between the predicted bound and the actual test error; smaller values indicate more accurate estimates of the discounted return prediction error. *Correlation*: We expect a linear correlation between the risk certificates and the observed test errors across policy instances.

**Computational requirements.** We conduct our experiments on a single computer equipped with a GeForce RTX 4090 GPU, an Intel(R) Core(TM) i7-14700K CPU (5.6 GHz), and 96 GB of memory. Training five policy instances to convergence in each environment takes about 30 minutes per instance, totaling 150 minutes. Collecting validation and test episodes requires around 20 minutes per policy level, or 60 minutes in total. Model training and PAC-Bayes bound computation across five policy instances, five repetitions, four baselines, and three policies takes four minutes per run, totaling roughly 1200 minutes per environment, 7000 minutes in total (about five days).

### 4.2 Results

We present full results on every environment, policy instance and repetition in Appendix D and restrict ourselves to discussing aggregated results in the main text.

**Strong correlation between bounds and test errors.** In Figure 2, we present scatter plots of all the PAC-Bayes bounds discussed in 4.1, policy instances, and repetitions against their respective test set errors across environments and levels of policy expertise. For every bound, the correlation between the bound and the test error increases with policy expertise. Within a fixed expertise level, the correlation also improves as the bound becomes more advanced, a trend that is already evident in more noisy *starter* policy. For example, in the brittle `Hopper` environment, which exhibits the weakest correlations overall, moving from NR-NI to R-I with T=6 raises the Pearson correlation from $0.4$ to $0.65$. At higher expertise levels, our recursive bounds achieve correlations above $0.9$ in almost all environments. Overall we see a clear linear trend, which demonstrates that our bounds are tight. There appears an increasing scatter as the expertise level decreases. This is expected, as the effects of an unconverged policy function on environment dynamics are less predictable. The bounds therefore provide a good prediction of the test-time return, answering Q1.

**Tightness improves with recursive depth.** In Figure 3 we plot the normalized bounds aggregated over policy instances and repetitions for each of the five environments. Smaller values reflect tighter bounds. Data-informed priors improve bounds across all environments for intermediate and expert policies, though this effect is less clear for the starter level policy. Introducing recursion (R-I, with T=2 and T=6) further tightens bounds, with deeper recursion generally yielding the tightest results. These improvements are most evident in environments with brittle dynamics such as `Humanoid` and `Hopper` where the locomotor has to keep its balance and less so in simpler environments such as `Half-Cheetah`. We see that while the correlation between bound and test-set error is already high, better, recursive, bounds provide improved tightness guarantees answering Q2.

**Recursion improves sample efficiency.** Collecting validation data from physical robots is often costly. Hence, the sample efficiency of a risk-certificate generation pipeline is of particular interest.

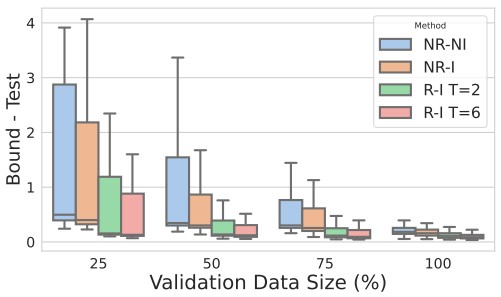 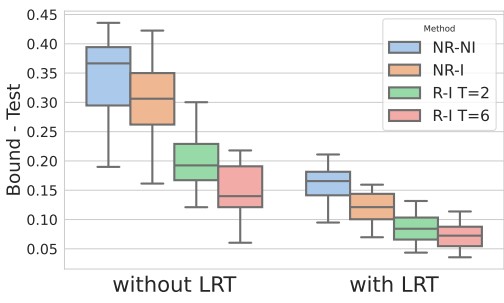

(a) The effect of validation data size on tightness

(b) The effect of local reparameterization on tightness.

Figure 4: *Bound tightness; smaller is better. Results are provided for the* `Humanoid` *environment, using five policy instances and five repetitions. (a) Tightness scores aggregated across three policy qualities and various validation set sizes, expressed as percentages of the full validation dataset. (b) Effect of the local reparameterization trick on bound tightness, illustrated for the expert-level policy.*

Figure 4a shows the tightness scores of the bounds across different validation data sizes in the `Humanoid` environment, while keeping the test size fixed. As expected, larger validation sets lead to tighter bounds, but the effect is most pronounced for our proposed recursive bounds. R-I T=6 achieves tightness results comparable to those that the non-recursive bounds (non-informed, NR-NI, and data-informed, NR-I) attain with twice as many data points. These findings demonstrate the ability of recursive bounds to significantly improve sample efficiency, addressing Q3.

**Local reparameterization improves tightness.** To train our model, we use a Bayesian neural network (BNN) that represents uncertainty by learning distributions over neural network parameters. To our knowledge, prior work on PAC-Bayesian risk certificate building with BNNs has relied exclusively on Blundell et al. [2015]'s *Bayes by backprop* approach [see, e.g., Pérez-Ortiz et al., 2021]. We show with Figure 4b that using the *local reparameterization trick (LRT)* [Kingma et al., 2015] to compute the empirical risk term in the bound calculation greatly improves bound tightness of all four evaluated bounds. This effects holds even in the already saturated expert-level policy of the challenging `Humanoid` environment. Further details can be found in Appendix D.

## 5   Limitations, future work, and broader impact

We restricted our empirical investigation to a single actor-critic algorithm and a single physics engine. This was a conscious choice to facilitate interpretation and maintain feasibility. Given the brittleness of the MuJoCo locomotion environments, we do not expect meaningful additional information to come from extending the same pipeline to RL suites with a similar level of fidelity. The next major step forward would be to implement our pipeline on a physical platform under controlled conditions. We considered only dense-reward locomotion scenarios with rigid locomotors, as this is the natural first step. The applicability of our findings to more advanced control settings, such as sparse-reward scenarios that require goal-conditioned or hierarchical RL algorithm design is subject to further investigation. We leave this enterprise to future work as the deep learning-based solutions for such setups have not yet reached the level of maturity to move beyond simulations. Another significant leap would be to proceed from our current self-certified policy evaluation approach to self-certified policy optimization in an online setting. This would necessitate training the policy via a PAC-Bayes bound. However, RL is a feedback-loop system in which assuring convergence, numerical stability, and optimal trade-offs between exploration and exploitation are major determinants of a stable training. While promising preliminary results exist [Tasdighi et al., 2024a,b], the problem is fundamental and requires a dedicated research program—an effort that goes beyond the scope of a single paper.

Our work contributes to the trustworthy development of agentic AI technologies, thereby promoting their adoption by society. Public concerns about such technologies will be even more pronounced when they are deployed on physical systems that are in direct contact with humans. Thanks to reliable risk certificates, such safety-critical technologies are likely to receive wider adoption. This, in turn, will further accelerate their development by expanding the pool of practice and observations.

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
