# OpenReview forum: "Deep Actor-Critics with Tight Risk Certificates"
_NeurIPS.cc/2025/Conference — Submitted to NeurIPS 2025_

### Official Review · Reviewer_ZwF5 · 2025-06-30

**Clarity:** 3
**Significance:** 3
**Originality:** 2
**Rating:** 5
**Confidence:** 3

**Summary:**

The paper addresses a critical problem for deploying deep actor-critic algorithms in physical systems: the absence of a validation scheme capable of quantifying their risk beforehand and this limits  their widespread adoption in physical world where hazards are real. The proposed solution involves developing tight risk certificates (in metric terms - a bound) for these algorithms (experiments are based on REDQ algorithm), which can accurately predict generalization performance based on validation-time observations. A key insight is the effectiveness of using minimal evaluation data, collected from a pretrained policy combined with a recursive PAC-Bayes approach. This method starts by fitting a
PAC-Bayes bound using observations from the validation set and then they predict the discounted return of the policy on the test set. Empirical results across multiple locomotion tasks demonstrate that these risk certificates are tight enough for practical use as they show correlation with test error. They also show that  tightness of bound improves as recursion depth increases. Crucially, the recursive bounds also significantly improve sample efficiency, making them particularly valuable for costly data collection in physical robot systems.

**Questions:**

I would like to know this for argument that requirement of  evaluation dataset is minimal. Can you tell validation dataset size as % of training dataset?

**Ethical Concerns:**

["NO or VERY MINOR ethics concerns only"]

**Final Justification:**

I am satisfied with the new eval results provided by authors.

**Limitations:**

Yes

**Quality:**

3

**Strengths And Weaknesses:**

Strengths :

1. Paper is well-written and easy to understand.
2. Addresses challenging problem of evaluating algorithms in RL in real world.
3. Seems fairly novel. Not aware of applying recursive PAC-Bayes in Deep actor critic algorithms, although have seen application of PAC-bayes in RL for model selection.
4. Good experimentation strategy. Specifically - a) Correlation between bound and test error for 3 levels of expertise. This can also serve as tool for convergence check, where noisy correlation between bound and true test error signifying lack of convergence. b) Results for different sizes of validation datasets.

Weaknesses :
1. Evaluated only off-policy type. Although authors do mention reason for choosing only one algorithm to delve deeper and perform more experiments rather shallow experiments on multiple ones but that said picking one from on-policy would have been great for e.g PPO which is right now go-to for LLMs.
2. [Open question for my understanding] Argument that requirement of evaluation dataset is minimal is not clear. It is written that model is trained for 3e5 steps, eval dataset is of 100 episodes. I would like to know how this dataset compares to training dataset in size in similar scale i.e, steps or maybe transition tuples.
3. Some good work in this track is missing in literature survey -
PAC-Bayesian Model Selection for Reinforcement Learning (Fard & Pineau, 2010)
PAC-Bayesian Policy Evaluation for Reinforcement Learning (Fard, Pineau & Szepesvári, 2012)

---

> ### Author Rebuttal · Authors · 2025-07-30
>
> Thanks for your in-depth review of our submission.
>
> ## Further evaluations
> We provide additional results on PPO and SAC on MuJoCo in our answer to MQz1.
> These results support that our findings for REDQ generalize to other RL models. Our recursive PAC-Bayesian bounds (R-I T=2; R-I T=6) provide tighter risk certificates than non-recursive bounds (NR-NI; NR-I), and the tightness increases with the recursive depth. See also our answer to em5X for further results as the recursive depth $T$ increases.
>
>
>
> To provide further evidence that they also generalize across benchmarks, we ran the same training and evaluation pipeline as reported in the paper on three environments each of the Meta-world (Yu et al., 2020) and DMC (Tassa et al., 2018) benchmarks.
> Our findings show the same performance pattern.
> The tables below show the tightness rank of each method for each environment averaged over seeds and policies.
> We will add the raw results, i.e., before averaging, to the camera-ready version of the paper.
>
> ##  MetaWorld -- REDQ (avg rank)
> | task        | method   |   train |   test |   last portion |
> |-------------|----------|------------------|-----------------|-------------------------|
> | drawer-open | NR-NI    |             3.4  |            3.4  |                  |
> | drawer-open | NR-I     |             2.27 |            2.17 |                  |
> | drawer-open | R-I T=2  |             2.23 |            2.27 |                    **1.47** |
> | drawer-open | R-I T=6  |             **2.1**  |            **2.17** |                    1.53 |
> | reach       | NR-NI    |             3.57 |            3.57 |                  |
> | reach       | NR-I     |             3.13 |            3.13 |                  |
> | reach       | R-I T=2  |             **1.63** |            **1.63** |                    **1.47** |
> | reach       | R-I T=6  |             1.67 |            1.67 |                    1.53 |
> | window-open | NR-NI    |             2.9  |            2.9  |                  |
> | window-open | NR-I     |             2.97 |            3    |                  |
> | window-open | R-I T=2  |             **2**    |            **2**    |                    **1.2**  |
> | window-open | R-I T=6  |             2.13 |            2.1  |                    1.8  |
>
>
> ##  DMControl -- REDQ (avg rank)
> | task                    | method   |   train |   test |   last portion |
> |-------------------------|----------|------------------|-----------------|-------------------------|
> | ball-in-cup-catch       | NR-NI    |             3.63 |            3.63 |                  |
> | ball-in-cup-catch       | NR-I     |             2.53 |            2.57 |                  |
> | ball-in-cup-catch       | R-I T=2  |             2.83 |            2.8  |                    2    |
> | ball-in-cup-catch       | R-I T=6  |             **1**    |            **1**    |                    **1**    |
> | reacher-hard | NR-NI    |             3.93 |            3.6  |                  |
> | reacher-hard | NR-I     |             2.37 |            2.53 |                  |
> | reacher-hard | R-I T=2  |             2.57 |            2.27 |                    1.93 |
> | reacher-hard | R-I T=6  |             **1.13** |            **1.6**  |                    **1.07** |
> | walker-run              | NR-NI    |             3.93 |            3.93 |                  |
> | walker-run              | NR-I     |             2.5  |            2.53 |                  |
> | walker-run              | R-I T=2  |             2.57 |            2.53 |                    2    |
> | walker-run              | R-I T=6  |             **1**    |            **1**    |                    **1**    |
>
>
>
> ## Clarify data set sizes
> During training the model is trained for 300k steps, i.e., it collects 300k transition tuples. The size of the evaluation data depends on the environments and the policy level. For example, in MuJoCo every episode has an upper limit of 1k transitions, but for more complicated environments, such as Hopper and Humanoid, they tend to end in failure before that, so collecting data for 100 episodes results in ~27k and ~47k respectively.
>
> In Figure 4 (a) we evaluate the influence of the evaluation data size. Note that while the overall spread decreases as the percentage of training data increases, as expected, the median tightness of our recursive bounds after 25 episodes is already almost identical to collecting 100 episodes worth of transitions.
>
>
> ## Literature survey lacks two references
> Fard & Pineau (2010) were the first to introduce a PAC-Bayesian bound in reinforcement learning, bounding the value function.
> However, their guarantees were limited to small discrete domains.
> Their follow-up work, Fard et al. (2012), extends this to value functions of fixed policies in a continuous domain.
> However, their bound assumes access to the exact environment model and can only handle a single-episode setting.
>
> We'll add this overview to the last paragraph in Appendix A, where we already cite Fard et al. (2012).
>
> _____
> Fard & Pineau (2010), _PAC-Bayesian Model Selection for Reinforcement Learning_  (NeurIPS)
> Fard et al. (2012), _PAC-Bayesian Policy Evaluation for Reinforcement Learning_ (UAI)
> Tassa et al. (2018), _Deepmind control suite_  (arXiv)
> Yu et al. (2020), _Meta-world: A benchmark and evaluation for multi-task and meta reinforcement learning._ (CoRL)

---

> > ### Comment · Reviewer_ZwF5 · 2025-08-05
> >
> > I am satisfied with authors’ reply and have updated my score accordingly.

---

### Official Review · Reviewer_MQz1 · 2025-07-03

**Clarity:** 3
**Significance:** 2
**Originality:** 2
**Rating:** 4
**Confidence:** 2

**Summary:**

The paper proposes a deployment-time certification method for deep, continuous-action actor-critics. After training a policy, ≈100 fresh roll-outs are recursively split to build increasingly informative PAC-Bayes priors, giving a closed-form upper bound on the trajectory-level gap from a target return. Bayes-by-Backprop with the local-reparameterisation trick supplies a low-variance posterior, yielding certificates that correlate (ρ > 0.9) with true returns on five MuJoCo tasks, esp at intermediate and expert skill levels.

**Questions:**

- Add at least one prior certificate as a baseline (e.g., PAC-Bayes SAC evaluated post-training or a discretised bound).
- Report sensitivity to deeper recursion and non-uniform split schedules; include wall-clock cost for posterior fitting + certificate computation.
- Demonstrate generality on a different algorithm (e.g., SAC) or a small experiment to show the method survives sim-to-real noise.
- Address safety tails: Either plot empirical return distributions or discuss how a CVaR-style loss could be substituted for the current return-gap term.

**Ethical Concerns:**

["NO or VERY MINOR ethics concerns only"]

**Final Justification:**

I have viewed the authors' rebuttal and it addresses some of my concerns.

**Limitations:**

The paper is forthright about simulation-only evidence and dependence on reward shaping, but should also note the bound’s blindness to rare catastrophic events and to online policy adaptation as it is done frequently in realistic settings.

**Paper Formatting Concerns:**

Typos:
- “Surprisingly, a small feasible of evaluation roll-outs collected “

**Quality:**

3

**Strengths And Weaknesses:**

Strengths:
- Novel evaluation setting: First work to post-facto certify deep actor-critics with so little validation data.
- Recursive prior refinement is a simple, theoretically sound trick that monotonically tightens bounds.
- Practical implementation: Adds minimal overhead to a standard REDQ pipeline; code is easy to reuse.
- Clear presentation: Figures link bound width to generalisation error; proofs are complete.

Weaknesses:
- Baseline availability is understated: Prior PAC-Bayes certificates exist for both continuous-action (PAC4SAC, PBAC, PAC) and discrete-action RL; at least one could be reported for fairness (discrete action ones could be trivially applied to this case by binning and checking for empirical tightness).
- Return-only guarantee. Bounding expected return ignores tail-risk, per-step safety constraints and reward-shaping artefacts, which could limit practical assurance.
- Single algorithm + simulator. No evidence on PPO/SAC or on real world robotics data or other control environments, only state space (no visual control) settings, so generality is unclear.

---

> ### Author Rebuttal · Authors · 2025-07-30
>
> We appreciate your detailed review of our submission.
>
> ## PAC4SAC, PBAC, and PAC
> PAC4SAC (Tasdighi et al., 2024) and PBAC (Tasdighi et al., 2025) are designed exclusively for policy optimization, using PAC-Bayesian principles as algorithmic guidelines rather than for reliable risk certificate generation, which is our objective. Both methods employ approximately calculated PAC-Bayesian bounds without quantifying the resulting approximation errors. While such approximations are acceptable in policy search contexts where learning performance is the main concern, they preclude making rigorous analytical statements about test-time performance and therefore cannot support risk certification.
>
> Specifically, PAC4SAC's bound includes an expected variance term (see Theorem 1 in Tasdighi et al., 2024) that requires perfect knowledge of transition dynamics, making it impractical for real-world applications. PBAC approximates the KL divergence computation (see Theorem 2 in Tasdighi et al., 2025), which is intractable for most nonlinear function spaces. Since these approximation errors remain uncharacterized, the resulting bounds cannot provide the reliability guarantees necessary for risk certificates.
>
> In contrast, our method is designed specifically for post-hoc risk certification, where bound tightness and theoretical guarantees are paramount. We will clarify this distinction in the first paragraph of Section 2.2 and Appendix A, where we cite these works.
>
> Regarding the PAC comment: Could the reviewer specify which particular model they are requesting? If they seek a generic non-recursive bound, we already include two non-recursive PAC-Bayesian bounds (NR-NI, NR-I) as baselines in our evaluation.
>
>
> ## Additional Environments and Baselines
> Upon your suggestion, we evaluated our risk certificate generation method on additional environments (MetaWorld) and baseline models (PPO, SAC). Due to runtime constraints, we restricted ourselves to three out of the original five environments.
> Below we report the rank for each task and method averaged over seeds and policies. The results support our findings with REDQ. More advanced bounds provide better risk certificates, with the most recursive version ($T=6$) dominating the others.
>
> See also results for REDQ on the Meta-world (Yu et al., 2020) and DMC (Tassa et al., 2018) benchmark environments in our answer to ZwF5.
> Our findings generalize across both environments and algorithms.
> We provide the full results and corresponding summary figures in the camera-ready version of the paper.
>
>
> ###  MuJoCo -- PPO (avg rank)
> | task     | method   |   train |   test |   last portion |
> |----------|----------|------------------|-----------------|-------------------------|
> | Cheetah  | NR-NI    |             3.44 |            3.44 |                     |
> | Cheetah  | NR-I     |             3    |            3    |                     |
> | Cheetah  | R-I T=2  |             2.56 |            2.56 |                       2 |
> | Cheetah  | R-I T=6  |             **1**    |            **1**    |                       **1** |
> | Hopper   | NR-NI    |             2.94 |            3    |                     |
> | Hopper   | NR-I     |             2.78 |            2.67 |                     |
> | Hopper   | R-I T=2  |             3.28 |            3.22 |                       2 |
> | Hopper   | R-I T=6  |             **1**    |            **1.11** |                       **1** |
> | Humanoid | NR-NI    |             3.78 |            3.83 |                     |
> | Humanoid | NR-I     |             2.33 |            2.33 |                     |
> | Humanoid | R-I T=2  |             2.89 |            2.83 |                       2 |
> | Humanoid | R-I T=6  |             **1**    |            **1**    |                       **1** |
>
> ###  MuJoCo -- SAC (avg rank)
> | task     | method   |   train |   test |   last portion |
> |----------|----------|------------------|-----------------|-------------------------|
> | Cheetah  | NR-NI    |             3.56 |            3.5  |                  |
> | Cheetah  | NR-I     |             3    |            3.06 |                  |
> | Cheetah  | R-I T=2  |             2.33 |            2.33 |                    1.94 |
> | Cheetah  | R-I T=6  |             **1.11** |            **1.11** |                    **1.06** |
> | Hopper   | NR-NI    |             3.72 |            3.67 |                  |
> | Hopper   | NR-I     |             2.33 |            2.33 |                  |
> | Hopper   | R-I T=2  |             2.39 |            2.33 |                    1.89 |
> | Hopper   | R-I T=6  |             **1.56** |            **1.67** |                    **1.11** |
> | Humanoid | NR-NI    |             3.44 |            3.44 |                  |
> | Humanoid | NR-I     |             3    |            2.89 |                  |
> | Humanoid | R-I T=2  |             2    |            2.06 |                    1.83 |
> | Humanoid | R-I T=6  |             **1.56** |            **1.61** |                    **1.17** |
>
> ## Empirical return distribution and CVaR-style loss
> Distributional RL (Bellemare et al., 2023) aims to model the complete return distribution and approximations stemming from projection and discretization errors, the uncertainties of which it cannot quantify completely. In contrast, our method accounts for all sources of uncertainty such as function approximation and finite sample effects, deriving probabilistic bounds with specified confidence levels that enable rigorous risk certificates. This distinction is crucial for safety-critical domains that require formal risk quantification with known confidence levels over learned distributional approximations without uncertainty quantification.
>
> Our results and discussion focus on risk certificates on the expected error, $\mathbb E[L(h)]$. However, the pipeline we introduce in Section 3 is agnostic to the chosen type of loss up to technical constraints, e.g., that it needs to be bounded. If $\text{CVaR}_\alpha(\pi)$ is the conditional value at risk for a policy $\pi$ and given confidence level $\alpha$, our approach can be directly adapted to provide a bound given an empirical estimate, $\text{CVaR}_\alpha(\hat \pi)$ (corresponding to $\hat L(h)$ in the terminology of our paper).
> The resulting risk certificate gives a tight bound on this risk-sensitive objective. A distributional approach would lack such formal guarantees.
>
>
> ## Deeper recursion and runtime
> Please see our runtime and recursion results in the answer to reviewer em5X.
> While increasing $T$ helps improve the tightness of the risk certificate, the improvements quickly become less pronounced. As the risk certificates are computed after training the model, which is the costly part, they add a negligible one-time post-training cost.
> See also the last paragraph in Section 4.1 for further runtime results.
>
> _____
> Bellemare et al. (2023), _Distributional Reinforcement Learning_  (MIT Press)
> Haarnoja et al. (2018), _Soft actor-critic algorithms and applications_ (ICML)
> Schulman et al. (2017), _Proximal policy optimization algorithms_  (arXiv)
> Tasdighi et al. (2024), _PAC-Bayesian soft actor-critic learning_  (AABI)
> Tasdighi et al. (2025), _Deep exploration with PAC-Bayes_  (arXiv)
> Tassa et al. (2018), _Deepmind control suite_  (arXiv)
> Yu et al. (2020), _Meta-world: A benchmark and evaluation for multi-task and meta reinforcement learning._ (CoRL)

---

### Official Review · Reviewer_em5X · 2025-07-03

**Clarity:** 3
**Significance:** 2
**Originality:** 2
**Rating:** 5
**Confidence:** 3

**Summary:**

This paper introduce a framwork for generating tight risk certificates for Deep AC RL using PAC-Bayes Bounds. auther demonstrates that even minimal evaluation data can accurate generalization guarantees.

**Questions:**

1. Please address the limitation 1 with better figure quality

2. From data efficency perspective, in order to achieve same quality of bound tightness, how much sample (in learning optimization process) the purposed method will save and how long clock time?

3. Will deeper T keep improve performance? what are bottleneck?

**Ethical Concerns:**

["NO or VERY MINOR ethics concerns only"]

**Final Justification:**

Auther promised they will revise the figure and they provided extra results to address my concerns.

**Limitations:**

yes

**Quality:**

3

**Strengths And Weaknesses:**

>Strength

1. paper is clear written.
2. auther did evaluate on varity of tasks.


>weakness

1. Figures are hard to see because of the dense information (Figure 2, 3,5 for some subfigures, the dense data make the result really hard to find). Espicially For figure 2, please have a legend on the figure. it especially hard to match the caption with the visual elements. For 3 and 5, in the appendix version, auther may plot it in other way to make it more clear.

2. The PAC-Bayes introduces a sequence of posteriors and scaling factors which may increase computational complexity.

3. Auther did study the quality of the learned policy. However the policy range is just from 100k to 300k which may be too limited espicially for complex task such as humanoid. Can auther report the evaluation quality of these learned policy?

---

> ### Author Rebuttal · Authors · 2025-07-30
>
> Thank you for taking the time to thoroughly review our submission.
>
> ## Changes to the visual representation
> - Figure 2: We will include a legend highlighting the color coding that is currently in the caption. We will also provide enlarged versions of these scatter plots in the appendix.
> - Box plots: These reflect the relative performance differences. We will rotate Figure 5 by 90 degrees and increase its size. We will also include a larger copy of Figure 3 with the same modification in the appendix.
>
> ## Runtime & Computational complexity
> The training time of the algorithm and the runtime of the learned policy when it is deployed are independent of the risk certificate computation.
> As such, there is no additional cost. The certificate computation itself is cheap and takes about a minute on a single GPU.
>
> For the expert-level policy on humanoid, the runtime cost in seconds averaged over three seeds as $T$ increases is roughly linear.
>
> |  2   | 4    |6     |8     |10    |
> |------|------|------|------|------|
> | 23.9 | 34.0 | 44.3 | 53.6 | 63.4 |
>
>
> See also the last paragraph of Section 4.1 (l276-282) for further runtime results.
>
> ## Data efficiency
> See Figure 4 (a) for an overview of the effect of validation data size on the bound tightness. Even with half the validation data the results are already close to the full validation data, with an almost identical median tightness given with as little as 25%.
>
> ## Deeper recursion
> The following tables provide results on increasing the recursive depth $T$ on MuJoCo Humanoid with REDQ and MetaWorld reach, each averaged over three seeds.
>
> While increasing the recursive depth $T$ tends to provide ever tighter risk certificates, the improvements quickly become negligible.
>
> ### Humanoid -- REDQ
>
> **Starter policy**
>
> |   T |   Train |   Test |   Bound |   Last |
> |-----|---------|--------|---------|--------|
> |   2 |   0.279 |  0.284 |   3.443 |  1.96  |
> |   4 |   0.19  |  0.184 |   3.39  |  1.179 |
> |   6 |   0.112 |  0.109 |   2.832 |  0.849 |
> |   8 |   0.081 |  0.079 |   2.147 |  1.01  |
> |  10 |   0.071 |  0.071 |   1.963 |  0.833 |
>
>
> **Intermediate policy**
>
> |   T |   Train |   Test |   Bound |   Last |
> |-----|---------|--------|---------|--------|
> |   2 |   0.1   |  0.102 |   0.348 |  0.257 |
> |   4 |   0.077 |  0.08  |   0.394 |  0.215 |
> |   6 |   0.07  |  0.073 |   0.342 |  0.183 |
> |   8 |   0.064 |  0.067 |   0.303 |  0.17  |
> |  10 |   0.064 |  0.066 |   0.295 |  0.181 |
>
>
> **Expert policy**
>
> |   T |   Train |   Test |   Bound |   Last |
> |-----|---------|--------|---------|--------|
> |   2 |   0.082 |  0.084 |   0.319 |  0.223 |
> |   4 |   0.075 |  0.078 |   0.31  |  0.183 |
> |   6 |   0.072 |  0.074 |   0.275 |  0.172 |
> |   8 |   0.07  |  0.071 |   0.287 |  0.168 |
> |  10 |   0.068 |  0.071 |   0.262 |  0.166 |
>
> ### MetaWorld Reach
>
> **Starter policy**
>
> |   T |   Train |   Test |   Bound |   Last |
> |-----|---------|--------|---------|--------|
> |   2 |   0.053 |  0.053 |   0.118 |  0.083 |
> |   4 |   0.05  |  0.05  |   0.127 |  0.078 |
> |   6 |   0.049 |  0.049 |   0.118 |  0.077 |
> |   8 |   0.048 |  0.048 |   0.104 |  0.074 |
> |  10 |   0.047 |  0.047 |   0.095 |  0.074 |
>
> **Intermediate policy**
>
> |   T |   Train |   Test |   Bound |   Last |
> |-----|---------|--------|---------|--------|
> |   2 |   0.082 |  0.081 |   0.097 |  0.092 |
> |   4 |   0.078 |  0.078 |   0.099 |  0.09  |
> |   6 |   0.079 |  0.079 |   0.101 |  0.091 |
> |   8 |   0.055 |  0.055 |   0.078 |  0.068 |
> |  10 |   0.076 |  0.076 |   0.091 |  0.081 |
>
> **Expert policy**
>
> |   T |   Train |   Test |   Bound |   Last |
> |-----|---------|--------|---------|--------|
> |   2 |   0.069 |  0.069 |   0.081 |  0.075 |
> |   4 |   0.048 |  0.048 |   0.069 |  0.058 |
> |   6 |   0.081 |  0.081 |   0.091 |  0.085 |
> |   8 |   0.022 |  0.022 |   0.042 |  0.032 |
> |  10 |   0.026 |  0.026 |   0.046 |  0.04  |
>
>
>
> ## Sufficiency of policy range evaluation
> See the reward curves of Chen et al. (2021). Figure 1 therein shows the performance of a learned policy over a range of 300k interaction steps on four of our five reported MuJoCo environments. The policies show significant performance differences after 100k, 200k, and 300k steps, respectively, and tend to converge toward the end of training.
> We'll mention this in the camera-ready version in line 254.
>
> _____
> Chen et al. (2021) _Randomized ensembled double Q-learning: Learning fast without a model_ (ICLR)

---

> > ### Comment · Reviewer_em5X · 2025-08-04
> >
> > Thanks auther for the rebuttal which address most of my concern. I will increase score to 5

---

### Official Review · Reviewer_fkS5 · 2025-07-03

**Clarity:** 3
**Significance:** 3
**Originality:** 2
**Rating:** 4
**Confidence:** 5

**Summary:**

This paper proposes a method for generating risk certificates for deep actor-critic architectures, aiming to provide high-confidence guarantees on generalization performance.

The method builds on the PAC-Bayes theoretical framework and introduces a recursive PAC-Bayes inference procedure. Traditional PAC-Bayes methods are difficult to apply directly to reinforcement learning due to their assumption of independent and identically distributed (i.i.d.) data, while reinforcement learning operates under a policy-driven Markov decision process. To address this challenge, the authors design a staged modeling and analysis process. The approach divides a small set of evaluation data into multiple subsets, sequentially trains a series of Bayesian neural networks, and recursively constructs upper bounds on generalization error at each stage. This strategy not only reduces the reliance on large validation datasets but also leverages prior training outcomes to improve the accuracy of risk estimation.

In the experimental section, the paper evaluates the method across five classic MuJoCo reinforcement learning environments—Ant, Hopper, Half-Cheetah, Humanoid, and Walker2d—covering different levels of policy quality (starter, intermediate, expert). The results show strong performance in multiple aspects: high confidence, tightness, sample efficiency and robustness.

In summary, the proposed method is theoretically sound and practically effective. It demonstrates strong applicability and scalability.

**Questions:**

See above.

**Ethical Concerns:**

["NO or VERY MINOR ethics concerns only"]

**Final Justification:**

This work proposes a method for generating risk certificates for deep actor-critic architectures, aiming to provide high-confidence guarantees on generalization performance.  Its experimental setup is well-designed, and it offers solid theoretical support.
However, despite the novelty of applying recursive PAC-Bayes theory to reinforcement learning tasks, the core theoretical techniques are mainly derived from existing work.  This limits the originality of the paper. After considering the response of the authors and the discussions from other reviewers, I decide to keep my score unchanged (borderline accept or weak accept).

**Limitations:**

Yes.

**Paper Formatting Concerns:**

None.

**Quality:**

3

**Strengths And Weaknesses:**

Strengths

1. The paper presents a carefully designed and well-executed experimental setup. It evaluates the proposed method across five diverse MuJoCo environments (Ant, Hopper, Half-Cheetah, Humanoid, and Walker2d), which are standard yet sufficiently complex benchmarks in continuous control. The experiments consider multiple policy quality levels (starter, intermediate, expert), account for randomness across multiple seeds and repetitions, and use appropriate statistical measures such as normalized error, correlation, and bound tightness. Additionally, the ablation studies assess the impact of recursion depth, validation data size, and local reparameterization tricks.  These experiments demonstrate not only the effectiveness but also the sample efficiency and robustness of the approach.
2. The paper provides strong theoretical support. The method is built on a solid foundation of PAC-Bayes theory, particularly the recent advances in recursive PAC-Bayes bounds. The authors clearly describe key theoretical concepts such as KL divergence, excess loss decomposition, and recursive bound construction.

Weaknesses

1. The Background section spans several pages and includes a deep dive into standard actor-critic reinforcement learning, PAC-Bayesian theory, and multiple versions of PAC-Bayes bounds. While these details are important, the extensive coverage overshadows the novel contributions of the paper. As a result, the structure feels unbalanced—the Background and Preliminaries occupy a disproportionate amount of space compared to the Methods and Results sections.
2. Although the application of recursive PAC-Bayesian bounds to reinforcement learning is novel in execution, the core theoretical techniques are largely drawn from prior work. The recursive PAC-Bayes method is not developed in this paper, and many of the theoretical tools (e.g., KL bounds, data-informed priors) have been well established in prior literature. This gives the impression that the paper mainly applies existing ideas to a new domain rather than proposing fundamentally new theory. As such, while the work is well-executed and useful, its originality may be viewed as incremental rather than groundbreaking.

---

> ### Author Rebuttal · Authors · 2025-07-30
>
> Thank you for your thorough review of our submission.
>
> ## Focus on the Background discussion and additional results
> We will move Section 2.2.1 to the appendix, while keeping Section 2.2.2 in the main text, as it provides the essential building block for the recursive bound, and place the focus on the current Section 2.2.3. We will use the resulting half-page of space to include figures and discussions of the additional experiments summarized in the answers to reviewers em5X, MQz1, and ZwF5.
> These experiments show that the results generalize to different models (PPO (Schulman et al., 2017) and SAC (Haarnoja et al., 2018)) and different benchmarks (Meta-world (Yu et al., 2020) and DMC (Tassa et al., 2018)).
>
> -----
> Haarnoja et al. (2018), _Soft actor-critic algorithms and applications_ (ICML)
> Schulman et al. (2017), _Proximal policy optimization algorithms_ (arXiv)
> Tassa et al. (2018), _Deepmind control suite_ (arXiv)
> Yu et al. (2020), _Meta-world: A benchmark and evaluation for multi-task and meta reinforcement learning._ (CoRL)

---

> > ### Comment · Reviewer_fkS5 · 2025-08-05
> >
> > Thank you for your reply. I will keep my score unchanged.

---

### Note · Authors · 2025-08-12

We thank the AC and the reviewers for granting us an active rebuttal period. We believe that all discussion threads have converged. We agree with reviewer MQz1 about the importance of assessing how our simulator findings correspond to physical robots. We hope this step will be taken in a follow-up study. We will acknowledge the limitation of our work due to a missing sim2real test in Section 5.

---

### Decision · Program_Chairs · 2025-09-17

**Decision:**

Reject

**Comment:**

The paper applies PAC-Bayes framework to actor-critic algorithms to obtain tight risk certificates in deep RL. The reviewers are generally positive towards the paper, appreciating the solid theory for the PAC-Bayes part and carefully performed experiments, the AC read the paper and find serious concerns in the overall framework and writing clarity, particularly in how the framework is being applied to RL; see detailed comments below. We are fully aware that rejection of a paper without negative reviews is unusual and should be exercised with caution, and the decision is made after the AC's careful reading of the paper and consultation with SAC and external reviewers.

---

**Correspondence between the PAC-Bayes framework and the actor-critic algorithm**: The paper applies PAC-Bayes framework to RL. However, the notation systems for PAC-Bayes and RL seem completely disconnected: PAC-Bayes part uses standard learning theory notation such as $L$ for loss, $h$ for hypothesis, $\rho\_0$ for prior, $D$ being dataset consisting of $(x,y)$ pairs, etc, whereas the RL part has policy $\pi$, critic $Q$, etc. **The paper does not connect these two notation systems.** As an example, what's hypothesis $h$ in the RL setting? Is it $\pi$ or $Q$? And what's loss? Critic loss (Section 2.1) or discounted return (Line 205)? The paper does not even address the mismatch in data format: in RL you get transition tuples $(s,a,r,s')$, and in supervised learning you get $(x,y)$. When you apply PAC-Bayes theory to RL, at least tell the readers what $x$ and $y$ are in the RL system. Section 4 has some text descriptions about the correspondence, yet the description is very vague and lacks precision. The AC asked this question to a reviewer and an external expert; despite both believe they understood the paper, their answers were different (and both different from the AC's guess). Some of us thinks $h$ is the policy and loss is the actor loss, some thinks $h$ is critic and loss is critic loss, and yet some thinks $h$ is a predictor that predicts Monte-Carlo return (rolled out by the trained policy) from state and loss is the Monte-Carlo prediction error.

This issue goes beyond the notation system. In PAC-Bayes, the final result is a guarantee on generalization error, which has well-understood meaning and applications in supervised learning. When you apply such "risk certificate" in RL, what is its meaning and how is it useful? In RL we care about the performance of the trained policy. Does the risk certificate provide a bound on it? If so, it should be stated formally in the RL notation system. Also, the closest concept to generalization error bounds in RL is finite-sample guarantees, which also involves quantities that address distribution shift (e.g., some notion of exploration rank/dimension or coverage parameter like boundedness of density ratio). If one starts with a generalization error bound for training (similar to those seen in supervised learning, say by uniform convergence in the simplest case), one needs to pay this additional factor about distribution shift to obtain the final guarantee. The lack of discussion on this raises further suspicion on what the guarantee in RL actually looks like.

---

**Description of TD**: The paper's description of TD in Line 87-90 is wrong. It claims that the algorithm minimizes the TD error which is an estimation of the Bellman error; this is a common misconception and technical mistake often made in casually written deep RL papers. TD loss requires freezing the target (no gradient is allowed to pass through Q(s',\pi)) and is _not_ an estimation of Bellman error, as the latter cannot be estimated from data (Sutton & Barto, 2nd ed, Chap 11.6) due to the infamous double-sampling difficulty. TD algorithm is not minimizing a single loss function but rather an iterative algorithm. If $h$ in the latter part of the paper is not critic, this issue will not have significant consequences to the results, but this is a sign that the authors are not rigorous about the RL part of their formulation despite being sophisticated about PAC-Bayes theory.

---

**Unbalanced Presentation**: As reviewer fkS5 pointed out,

> The Background section spans several pages and includes a deep dive into standard actor-critic reinforcement learning, PAC-Bayesian theory, and multiple versions of PAC-Bayes bounds. While these details are important, the extensive coverage overshadows the novel contributions of the paper.

In particular, pages 4 and 5 (which usually correspond to the core methodological developments in a paper) are dedicated to a modest adaptation and extension of existing PAC-Bayes bound, which has almost nothing to do with RL despite the title suggests an RL-focused paper. I see two options here: (1) if the authors want to keep the title, which implies an RL-focused paper, pages 4 and 5 should be drastically shorten with most derivations pushed to appendices, leaving more space to elaborate carefully how it is applied to RL. (2) if the authors want to keep the structure of the main text, the title should be changed to something like "Recursive PAC-Bayes Bounds for Continuous Losses, with Applications in Reinforcement Learning".